# Beyond Membership: Limitations of Add / Remove Adjacency in Differential Privacy

**Gauri Pradhan**
University of Helsinki, Finland
gauri.pradhan@helsinki.fi

**Joonas Jälkö**
University of Helsinki, Finland
joonas.jalko@helsinki.fi

**Santiago Zanella-Bèguelin**
Microsoft, Cambridge, UK
santiago@microsoft.com

**Antti Honkela**
University of Helsinki, Finland
antti.honkela@helsinki.fi

## Abstract

Training machine learning models with differential privacy (DP) limits an adversary's ability to infer sensitive information about the training data. It can be interpreted as a bound on adversary's capability to distinguish two adjacent datasets according to chosen adjacency relation. In practice, most DP implementations use the add/remove adjacency relation, where two datasets are adjacent if one can be obtained from the other by adding or removing a single record, thereby protecting membership. In many ML applications, however, the goal is to protect attributes of individual records (e.g., labels used in supervised fine-tuning). We show that privacy accounting under add/remove overstates attribute privacy compared to accounting under the substitute adjacency relation, which permits substituting one record. To demonstrate this gap, we develop novel attacks to audit DP under substitute adjacency, and show empirically that audit results are inconsistent with DP guarantees reported under add/remove, yet remain consistent with the budget accounted under the substitute adjacency relation. Our results highlight that the choice of adjacency when reporting DP guarantees is critical when the protection target is per-record attributes rather than membership.

## 1 Introduction

Differential Privacy (DP) (Dwork et al., 2006) provides provable protection against the most common privacy attacks, including membership inference, attribute inference and data reconstruction (Salem et al., 2023). It limits an adversary's ability to distinguish between two adjacent datasets based on the an algorithm's output. The level of DP guarantee depends on the underlying adjacency relation. There exist different notions of adjacency such as the *add/remove* adjacency, where two datasets differ by the inclusion or removal of a single record. An alternative is *substitute* adjacency, where one dataset is obtained by replacing a record in the other. A special case of the latter is *zero-out* adjacency, in which a record is replaced with a null entry. In deep learning (Abadi et al., 2016; Ponomareva et al., 2023), the standard approach to DP uses add/remove adjacency, that was designed to protect against an adversary's ability to detect whether an individual was part of the training dataset or not.

In this paper, we draw attention to the fact that while DP can provide protection against all the common attacks listed above, the add/remove adjacency does not provide protection against inference attacks on data of a subject known to be a part of the training dataset at the level indicated by the privacy parameters. Protection against such inference attacks requires considering substitute adjacency, which protects against inference of a single individual's contribution to the data. An add/remove privacy bound implies a substitute privacy bound, but with substantially weaker privacy parameters. Most DP libraries (such as Opacus Yousefpour et al. (2021)) implement privacy accounting assuming add/remove adjacency. A practitioner concerned with attribute or label privacy who relies on these libraries to train their model with DP may therefore be misled: the guarantees provided by add/remove adjacency overstate the actual protection against attribute inference attacks.

In order to evaluate practical vulnerability of DP models and mechanisms to substitute-type attacks, we develop a range of auditing tools for the substitute adjacency and apply these to DP deep learning. In this setting, we craft a pair of neighbouring datasets, $\mathcal{D}$ and $\mathcal{D}'$ by replacing a target record $z \in \mathcal{D}$ with a canary record $z'$. A canary serves as a probe that enables the adversary to determine whether a model was trained on $\mathcal{D}$ or $\mathcal{D}'$. We find that the algorithms do indeed leak more information to a training data inference attacker than the add/remove bound would suggest.

**Our Contributions:**

- We propose algorithms for crafting canaries for auditing DP under substitute adjacency, providing tight empirical lower bounds matching theoretical guarantees from accountants (Section 3).
- We show that privacy leakage can exceed the guarantees derived from add/remove accountants but (as expected), closely tracks the guarantees predicted by substitute accountants (Section 6).
- Our results demonstrate that accounting for privacy under the commonly used add/remove adjacency overstates the protection against attribute inference, including label inference.

## 2    RELATED WORK AND PRELIMINARIES

### 2.1    DIFFERENTIAL PRIVACY

Differential Privacy (DP) (Dwork et al., 2006) is a framework to protect sensitive data used for data analysis with provable privacy guarantees.

**Definition 1** (($\varepsilon, \delta, \sim$)-Differential Privacy). *A randomized algorithm $\mathcal{M}$ is $(\varepsilon, \delta, \sim)$-differentially private if for all pairs of adjacent datasets $\mathcal{D} \sim \mathcal{D}'$, and for all events $S$:*

$$\Pr[\mathcal{M}(\mathcal{D}) \in S] \leq e^{\varepsilon} \Pr[\mathcal{M}(\mathcal{D}') \in S] + \delta,$$

Under add/remove adjacency ($\sim_{AR}$), $\mathcal{D}'$ is obtained by adding or removing a record $z$ from $\mathcal{D}$. In substitute adjacency ($\sim_S$), $\mathcal{D}'$ is formed by replacing a record $z$ in $\mathcal{D}$ with another record $z'$. Kairouz et al. (2021) also introduced the zero-out adjacency which corresponds to removing a record from $\mathcal{D}$ and replacing it with a zero-out record ($\perp$) to form $\mathcal{D}'$. Privacy guarantees for this adjacency are semantically equivalent to the add/remove DP.

### 2.2    DIFFERENTIALLY PRIVATE STOCHASTIC GRADIENT DESCENT (DP-SGD)

Differentially Private Stochastic Gradient Descent (DP-SGD) (Rajkumar & Agarwal, 2012; Song et al., 2013; Abadi et al., 2016) forms the basis of training machine learning algorithms with DP. It is used to train ML models while satisfying DP. Given a minibatch $B_t \in \mathcal{D}$ at time step $t$, DP-SGD first clips the gradients for each sample in $B_t$ such that the $\ell_2$ norm for per-sample gradients does not exceed the clipping bound $C$. Following that, Gaussian noise with scale $\sigma C$ is added to the clipped gradients. These clipped and noisy gradients are then used to update the model parameters $\theta$ during training as follows:

$$\theta_{t+1} \leftarrow \theta_t - \frac{\eta}{|B|} \Big[ \sum_{z \in B_t} \texttt{clip}(\nabla_\theta \ell(\theta_t; z), C) + Z_t \Big], \tag{1}$$

where $Z_t \sim \mathcal{N}(0, \sigma^2 C^2 \mathbb{I})$, $|B|$ is the expected batch size, and $\eta$ denotes the learning rate of the training algorithm. In this way, DP-SGD bounds the contribution of an individual sample to train the model. In this paper, we also use DP-Adam which is the differentially private version of the Adam (Kingma & Ba, 2015) optimizer.

DP provides upper bounds for the privacy loss expected from an algorithm for a given adjacency relation. Early works used advanced composition (Dwork et al., 2010; Kairouz et al., 2015) to account for the cumulative privacy loss over multiple runs of a DP algorithm. Abadi et al. (2016); Mironov (2017); Bun & Steinke (2016) developed accounting methods for deep learning algorithms. However, the bounds on DP parameters provided by these accountants are not always tight. Recently, numerical accountants based on privacy loss random variables (PRVs) (Dwork & Rothblum, 2016; Meiser & Mohammadi, 2018) have been adopted across industry and academia (Koskela et al., 2020; Gopi et al., 2021) because they offer tighter estimates of DP upper bounds.

## 2.3 AUDITING DIFFERENTIAL PRIVACY

Privacy auditing helps evaluate the empirical privacy leakage from a differentially private machine learning algorithm. DP auditing involves assessing the privacy it affords to worst-case canary records. Jayaraman & Evans (2019) were the first to evaluate the empirical privacy leakage from machine learning models trained with DP-SGD and revealed a large gap between the empirical leakage and the theoretical bounds guaranteed by DP-SGD. Later, Nasr et al. (2021) audited DP machine learning algorithms under progressively stronger threat models. They show that the empirical privacy leakage from their strongest threat model using worst-case dataset canaries was "tight" with respect to the privacy accounting upper bound for DP. Subsequent works such as Nasr et al. (2023); Steinke et al. (2023); Annamalai & Cristofaro (2024); Zanella-Béguelin et al. (2023); Mahloujifar et al. (2025); Cebere et al. (2025) have since been focused on crafting worst-case canary records that could yield tight auditing for models trained with natural datasets with the more recent works focusing on practical threat models.

Threat models in auditing differ by the adversary's level of access: in the *White-Box* setting, the adversary can access the intermediate models during training (Nasr et al., 2021; 2023; Steinke et al., 2023); in the more realistic *Hidden-State* setting, the adversary can only access the final model but may still perturb inputs to intermediate models (Annamalai, 2024; Cebere et al., 2025); and in the *Black-Box* setting (Annamalai & Cristofaro, 2024; Boglioni et al., 2025), the adversary can only insert canary sample(s) at the start of training and tracks the final trained model's response on these canary sample(s).

## 3 AUDITING DP WITH SUBSTITUTE ADJACENCY

Our goal is to design canary samples for auditing DP under substitute adjacency in a *hidden-state* threat model. In this setting, the adversary can only access the final model at any step $t$, without visibility into prior intermediate models. Table 1 briefly describes the crafting scenarios for canaries used to audit DP with substitute adjacency. In Table 2, we detail the adversary's prior knowledge in each scenario. Algorithm 1 presents the method to audit DP in a substitute-adjacency threat model.

### 3.1 AUDITING MODELS USING CRAFTED WORST-CASE DATASET CANARIES

DP gives an upper bound on privacy loss of an algorithm. It assumes that the adversary can access the gradients from the mechanism. Furthermore, it guarantees that the privacy of a target record (crafted to yield worst-case gradient) holds even when the adversary constructs a worst-case pair of neighbouring datasets $(\mathcal{D}, \mathcal{D}')$. Thus, any privacy

---

**Algorithm 1** Privacy Auditing With Substitute Adjacency

**Requires:** Model Architecture $\mathbb{M}$, Model Initialization $\theta_0$, Dataset $\mathcal{D}$, Target Sample $z$, Training Loss $\ell$, Training Steps $T$, learning rate $\eta$, Optimizer $\mathtt{opt\_step}()$, Crafting Algorithm $\mathtt{craft}()$, DP Parameters $(\sigma, C, q)$, Repeats $R$, Crafting $\in$ {Gradient-Space, Input-Space}.

1: $\mathcal{O} \leftarrow \mathbf{0}_R, \mathcal{B} \leftarrow \mathbf{0}_R$
   ▷ **Adversary as Crafter:**
2: **if** Crafting = Gradient-Space **then**
3:   $g_z, g_{z'} \leftarrow \mathtt{craft}(\mathbb{M}, \mathcal{D}, \theta_0, T, \eta, \ell, C, q, \mathtt{opt\_step})$
4: **else**
5:   $z' \leftarrow \mathtt{craft}(z, \mathbb{M}, \mathcal{D}, \theta_0, T, \eta, \ell, \mathtt{opt\_step})$
6: **for** $r \in 1, ..., R$ **do**
   ▷ **Challenger as Model Trainer:**
7:   Choose $b$ uniformly at random: $b \sim \{0, 1\}$
8:   $\mathcal{B}[r] \leftarrow b$
9:   **for** $t \in 1, ..., T$ **do**
10:     Sample $B_t$ from $\mathcal{D}$ with prob. $q$
11:     $g_{\theta_t} \leftarrow \mathbf{0}_{|\theta|}$
12:     **for** $z_i \in B_t$ **do**
13:       $g_{\theta_t} \leftarrow g_{\theta_t} + \mathtt{clip}(\nabla_\theta(l(z_i)), C)$
14:     **if** b = 0 **then**
15:       $g_{\theta_t} \leftarrow g_{\theta_t} + [\mathtt{clip}(\nabla_\theta(\ell(\theta_t; z)), C) \text{ or } +g_z]$ with prob. $q$
16:     **else**
17:       $g_{\theta_t} \leftarrow g_{\theta_t} + [\mathtt{clip}(\nabla_\theta(\ell(\theta_t; z')), C) \text{ or } +g_{z'}]$ with prob. $q$
18:     $g_{\theta_t} \leftarrow g_{\theta_t} + \mathcal{N}(0, \sigma^2 C^2 \mathbb{I})$
19:     $\theta_{t+1} \leftarrow \mathtt{opt\_step}(\theta_t, g_{\theta_t}, \eta)$
   ▷ **Adversary as Distinguisher:**
20:   $\mathcal{O}[r] \leftarrow \mathtt{logit}(z; \theta_T) - \mathtt{logit}(z'; \theta_T) \text{ or } \left(\frac{g_z}{C}\right) \cdot (\theta_T - \theta_0)$
21: **return** $\mathcal{O}, \mathcal{B}$

---

auditing procedure with such a strong adversary yields tightest empirical lower bound on privacy parameters. Nasr et al. (2021) were the first to propose an auditing procedure which is provably tight for worst-case neighbouring datasets crafted to audit DP with add/remove adjacency.

Table 1: Crafting schema for auditing privacy leakage under substitute adjacency with varying adversary capabilities. The adversary can either craft canaries that allow them to directly manipulating the gradient input to the DP algorithm or they are restricted to input-space perturbations to craft the canary samples.The adversary's visibility into the training process is defined by the following threat models: (a) Visible-State (commonly known in the literature as White-Box), where the adversary assumes access to gradients from the model, and (b) Hidden-State, where they rely on model parameter updates/ output logits to estimate privacy loss.

| Scenario | Crafting Space | Type of Canary | Crafting Algorithm | Distinguishability Score | Threat Model |
|---|---|---|---|---|---|
| S1 | Gradient | Crafted Dataset | Section 3.1 | $\log(\Pr(g_T\|\mathcal{D})) - \log(\Pr(g_T\|\mathcal{D}'))$ | Visible-State |
| S2 | Gradient | Crafted Gradient | Algorithm 2 | $\theta_T - \theta_0$ | Hidden-State |
| S3 | Input | Crafted Input Sample | Algorithm 3 | $\texttt{logit}(z;\theta_T) - \texttt{logit}(z';\theta_T)$ | Hidden-State |
| S4 | Input | Crafted Mislabeled Sample | Algorithm 4 | $\texttt{logit}(z;\theta_T) - \texttt{logit}(z';\theta_T)$ | Hidden-State |
| S5 | Input | Adversarial Natural Sample | Algorithm 5 | $\texttt{logit}(z;\theta_T) - \texttt{logit}(z';\theta_T)$ | Hidden-State |

We craft $\mathcal{D}$ and $\mathcal{D}'$ as worst-case neighbouring datasets under substitute adjacency (scenario S1 in Table 1). Assuming $\mathcal{D}$ has a sample $z$ which yields a gradient $g_z$ such that $\|g_z\| = C$ throughout training. For maximum distinguishability, we form $\mathcal{D}'$ by replacing $z$ with $z'$ such that $\|g_{z'}\| = C$ but it is directionally opposite to $g_z$. For all the other samples in $\mathcal{D}$ and $\mathcal{D}'$, we assume that they contribute 0 gradients during training. Unlike Nasr et al. (2021), we do not assume that the learning rate is 0 for the steps with no gradient canary in the minibatch

Table 2: Adversary's prior knowledge in each auditing scenario described in Table 1.

| Priors | Scenario | | | | |
|---|---|---|---|---|---|
| | S1 | S2 | S3 | S4 | S5 |
| **Data Distribution** | − | ✓ | ✓ | ✓ | ✓ |
| **Target Sample ($z$)** | − | − | ✓ | ✓ | ✓ |
| **Model Architecture** | ✓ | ✓ | ✓ | ✓ | ✓ |
| **Training Hyperparameters** | − | ✓ | ✓ | ✓ | ✓ |
| **Subsampling Rate ($q$)** | − | ✓ | ✓ | ✓ | ✓ |
| **Clipping Bound ($C$)** | ✓ | ✓ | − | − | − |
| **Noise Multiplier ($\sigma$)** | − | − | − | − | − |

since this discounts the effect of subsampling on auditing. Since we account for the noise contribution by the minibatches without $z$ or $z'$, our setting more accurately reflects the true dynamics of DP-SGD. We further assume the adversary cannot access intermediate updates and observes only the final gradients from the mechanism.

At any step $T$, given subsampling rate $q$, the number of times the canary is sampled over $T$ steps is a binomial, $\mathcal{B} \sim \text{Binomial}(T, q)$. Conditioned on $\mathcal{B} = k$, the cumulative gradient $g_T$ given by

$$\Pr(g_T|\mathcal{B} = k) \sim \mathcal{N}(\pm kC, T\sigma^2 C^2). \tag{2}$$

The marginal distribution of $g_T$ over $\mathcal{D}$ or $\mathcal{D}'$ at step $T$ is given by

$$\Pr(g_T|\mathcal{D} \text{ or } \mathcal{D}') = \sum_{k=0}^{T} \binom{T}{k} q^k (1-q)^{T-k} \mathcal{N}(g_T; \pm kC, T\sigma^2 C^2), \tag{3}$$

where $C$ is the gradient contribution of $\mathcal{D}$ and $-C$ of $\mathcal{D}'$. The adversary can use Equation (3) to compute $\texttt{log}(\Pr(g_T|\mathcal{D})) - \texttt{log}(\Pr(g_T|\mathcal{D}'))$ as the scores to compute the empirical lower bound for $\varepsilon_S$ during auditing.

## 3.2 Auditing Models Trained With Natural Datasets

While DP offers protection to training samples against worst-case adversaries, high-utility ML models are obtained by training on natural datasets. Under substitute adjacency, $\mathcal{D}$ and $\mathcal{D}'$ differ by replacing a target sample $z$ in $\mathcal{D}$ with $z'$. Effective auditing for models trained with natural datasets, therefore requires canaries that maximize the distinguishability between the two datasets.

### 3.2.1 Crafting Canaries For Auditing In Gradient Space

Recently, Cebere et al. (2025) propose a worst-case gradient canary for tight auditing on models trained with add/remove DP using natural datasets in a hidden state threat model. Adapting their idea to substitute adjacency-based auditing, we first select the trainable model parameter which changes least in terms of its magnitude throughout training.

We then define canary gradients $g_z$ and $g_{z'}$ by setting all other parameter gradients to $0$, and assigning a magnitude $C$ to the gradient of the selected least-updated parameter. This ensures that $\|g_z\| = \|g_{z'}\| = C$. For maximum distinguishability between $g_z$ and $g_{z'}$, we orient them in opposite directions in gradient space. The detailed procedure for constructing these canaries is provided in Algorithm 2. For computing the empirical privacy leakage, we record change in parameter from initialization, $\theta_t - \theta_0$ as scores for auditing. These scores serve as proxies for the adversary's confidence that the observed outputs were from model trained on $\mathcal{D}$ or $\mathcal{D}'$. This setting corresponds to scenario S2 in Table 1. Such canaries can be used to audit models trained using federated learning.

---

**Algorithm 2** Generating Crafted Gradient Canary Pair $(g_z, g_{z'})$

---

**Requires:** Dataset $\mathcal{D}$, Training Loss $\ell$, Model Initialization $\theta_0$, Training Steps $T$, Learning Rate $\eta$, Clipping Bound $C$, Optimizer `opt_step()`.

1: **def** `craft`:
2:    $S \leftarrow \mathbf{0}_d$ s.t. $d \leftarrow |\theta_0|$
3:    **for** $t \in 1, ..., T$ **do**
4:      Sample $B_t$ from $\mathcal{D}$
5:      $\overline{g}_{\theta_t} \leftarrow \texttt{clip}(\nabla_\theta \ell(\theta_t; z_i), C)$
6:      $\theta_{t+1} \leftarrow \texttt{opt\_step}(\theta_t, \overline{g}_{\theta_t}, \eta)$
7:      **for** $j \in 1, ..., d$ **do**
8:        $S_j \leftarrow S_j + \left|\theta_{t+1}^j - \theta_t^j\right|$
9:    $j^* \leftarrow \texttt{argmin}_{1 \le j \le d}(S_j)$
10:   $g_z \leftarrow \mathbf{0}_d$
11:   $g_z[j^*] \leftarrow C$
12:   $g_{z'} \leftarrow \mathbf{0}_d$
13:   $g_{z'}[j^*] \leftarrow -C$
14:   **return** $g_z, g_{z'}$

---

### 3.2.2 CRAFTING CANARIES FOR AUDITING IN INPUT SPACE

In practice, adversaries are unlikely to directly manipulate a model's gradient space during training. In such cases, the adversary is constrained to input-space perturbations where a natural sample $z \in \mathcal{D}$ will be replaced with an adversarially crafted sample $z'$ to form $\mathcal{D}'$ prior to training. For instance, an adversary could mount a data-poisoning attack during the fine-tuning of a large model, or attempt to infer the label of a known-in-training user. For input-space canaries, we track $\texttt{logit}(z; \theta_t) - \texttt{logit}(z'; \theta_t)$ as scores for auditing.

For auditing using input-space canaries, we begin by selecting a target sample ($z$) for which the a reference model (trained without DP) exhibits least-confidence over training. The crafted canary equivalent ($z'$) can then be generated using the following criteria:

---

**Algorithm 3** Generating Crafted Input Canary $(z' \sim (x', y))$

---

**Requires:** Target Sample $z \sim (x, y)$, Dataset $\mathcal{D}$, Training Loss $\ell$, Model $\mathbb{M}$, Model Initialization $\theta_0$, Training Steps $T$, Crafting Steps $N$, Learning Rate $\eta$.

1: **def** `craft`:
2:    $\theta_T \leftarrow \texttt{train}(\mathbb{M}, \theta_0, \mathcal{D}, T, \ell, \eta)$
3:    $z' \sim (x', y)$ s.t. $x' \leftarrow \mathbf{0}_{|x|}$
4:    $\mathcal{L}_{\text{cosim}}(x') \leftarrow \dfrac{\nabla_\theta \ell(\theta_T; x, y) \cdot \nabla_\theta \ell(\theta_T; x', y)}{\|\nabla_\theta \ell(\theta_T; x, y)\| \cdot \|\nabla_\theta \ell(\theta_T; x', y)\|}$
5:    $\mathcal{L}_{\text{MSE}}(x') \leftarrow \text{MSE}(\nabla_\theta \ell(\theta_T; x, y), \nabla_\theta \ell(\theta_T; x', y))$
6:    **for** $n \in 1, ..., N$ **do**
7:      $x' \leftarrow x' - \eta(\nabla\mathcal{L}_{\text{cosim}}(x') + \nabla\mathcal{L}_{\text{MSE}}(x'))$
8:    **return** $z'$

---

- Algorithm 3 is used to generate a *crafted input* canary $z' \sim (x', y)$ complementary to the target sample $z$ (Scenario S3 in Table 1). It uses the reference model to craft $z'$ such that the cosine similarity between $g_z$ and $g_{z'}$ is minimized while ensuring that $g_{z'}$ is similar in scale to $g_z$ so that the model interprets $z'$ as a legitimate sample from the data distribution.

- Algorithm 4 is used to generate a *crafted mislabeled* canary $z' \sim (x, y')$ complementary to the target sample $z$ (Scenario S4 in Table 1). We use the reference model to find a label $y'$ in the label space $\mathcal{Y}$ such that it minimizes cosine similarity between $g_{z'}$ and $g_{z'}$.
- Algorithm 5 is used to select an *adversarial natural* canary $z' \sim (x', y')$ from an auxiliary dataset $\mathcal{D}_{\text{aux}}$ (formed using a subset of samples not used for training the model) complementary to the target sample $z$ (Scenario S5 in Table 1). We use the reference model to find a sample $z'$ in $\mathcal{D}_{\text{aux}}$ which yields minimum cosine similarity between $g_{z'}$ and $g_{z'}$.

## 4 USE OF GROUP PRIVACY TO APPROXIMATE SUBSTITUTE ADJACENCY YIELDS SUBOPTIMAL UPPER BOUNDS

By the definition of DP with substitute adjacency (Definition 1), $\mathcal{D}'$ can be obtained from $\mathcal{D}$ by removing a record $z$ and adding another record $z'$ to $\mathcal{D}$. As such, it is a common practice to infer

Substitute adjacency as a composition of one Add and one Remove operation (Kulesza et al., 2024). According to Dwork & Roth (2014), if an algorithm $\mathcal{M}$ satisfies $(\varepsilon, \delta, \sim_{AR})$-DP, then for any pair of $\mathcal{D}$ and $\mathcal{D}'$ that differ in at most $k$ records, the following relationship holds true

$$\Pr[\mathcal{M}(\mathcal{D}) \in S] \leq e^{k\varepsilon} \Pr[\mathcal{M}(\mathcal{D}') \in S] + \Big(\sum_{i=0}^{k-1} e^{i\varepsilon}\Big)\delta. \tag{4}$$

From Equation (4), it follows that

**Theorem 4.1** (Dwork & Roth (2014)). *Any algorithm $\mathcal{M}$ which satisfies $(\varepsilon_{AR}, \delta_{AR}, \sim_{AR})$-DP is $(\varepsilon_S, \delta_S, \sim_S)$-DP with $\varepsilon_S = 2\varepsilon_{AR}$ and $\delta_S = (1 + e^{\varepsilon_{AR}})\delta_{AR}$.*

Theorem 4.1 yields an upper bound for substitute DP derived from add/remove DP which is agnostic of the underlying algorithm. For certain algorithms (such as the Poisson-subsampled DP-SGD used in this paper), which can be characterized by privacy loss random variables (PRVs) and their corresponding privacy loss distribution (PLD) (Dwork & Rothblum, 2016; Meiser & Mohammadi, 2018; Koskela et al., 2020), numerical accountants can derive the privacy curve directly. This approach is recommended over using general, algorithm-agnostic upper bounds, as it provides significantly tighter privacy guarantees. Moreover, Theorem 4.1 assumes scaled $\delta$; with fixed $\delta$, $\varepsilon_S$ may exceed $\varepsilon_{AR}$ (as shown in Figure A5, Appendix A.3)

---

**Algorithm 4** Generating Crafted Mislabeled Canary ($z' \sim (x, y')$)

**Requires:** Target Sample $z \sim (x, y)$, Dataset $\mathcal{D}$, Training Loss $\ell$, Model $\mathbb{M}$, Model Initialization $\theta_0$, Training Steps $T$, Learning Rate $\eta$, Label Space $\mathcal{Y}$.

1: **def** craft:
2:   $\theta_T \leftarrow \texttt{train}(\mathbb{M}, \theta_0, \mathcal{D}, T, \ell, \eta)$
3:   $S \leftarrow \mathbf{0}_d$ s.t. $d \leftarrow |\mathcal{Y}|$
4:   **for** $\hat{y} \in \mathcal{Y}$ **do**
5:     $\hat{z} \sim (x, \hat{y})$
6:     $S[\hat{y}] \leftarrow \dfrac{\nabla_\theta \ell(\theta_T; z) \nabla_\theta \ell(\theta_T; \hat{z})}{\|\nabla_\theta \ell(\theta_T; z)\| \|\nabla_\theta \ell(\theta_T; \hat{z})\|}$
7:   $j^* \leftarrow \texttt{argmin}_{1 \leq j \leq d}(S_j)$
8:   $y' \leftarrow \mathcal{Y}[j^*]$
9:   **return** $z'$

**Algorithm 5** Selecting Canary From Natural Samples($z' \sim (x', y')$)

**Requires:** Target Sample $z \sim (x, y)$, Dataset $\mathcal{D}$, Training Loss $\ell$, Model $\mathbb{M}$, Model Initialization $\theta_0$, Training Steps $T$, Learning Rate $\eta$, Auxiliary Dataset $\mathcal{D}_{\text{aux}}$.

1: **def** craft:
2:   $\theta_T \leftarrow \texttt{train}(\mathbb{M}, \theta_0, \mathcal{D}, T, \ell, \eta)$
3:   $S \leftarrow \mathbf{0}_d$ s.t. $d \leftarrow |\mathcal{D}_{\text{aux}}|$
4:   **for** $\hat{z} \in \mathcal{D}_{\text{aux}}$ **do**
5:     $\hat{z} \sim (\hat{x}, \hat{y})$
6:     $S[\hat{z}] \leftarrow \dfrac{\nabla_\theta \ell(\theta_T; z) \nabla_\theta \ell(\theta_T; \hat{z})}{\|\nabla_\theta \ell(\theta_T; z)\| \|\nabla_\theta \ell(\theta_T; \hat{z})\|}$
7:   $j^* \leftarrow \texttt{argmin}_{1 \leq j \leq d}(S_j)$
8:   $z' \leftarrow \mathcal{D}_{\text{aux}}[j^*]$
9:   **return** $z'$

---

## 5 GENERAL EXPERIMENTAL SETTINGS

**Training Details:**

- **Training Paradigm**: We fine-tune the final layer of ViT-B-16 (Dosovitskiy et al., 2021) model pretrained on ImageNet21K. We also fine-tune a linear layer on top of Sentence-BERT (Reimers & Gurevych, 2019) encoder for text classification experiments. We use a 3-layer fully-connected multi-layer perceptron (MLP) (Shokri et al., 2017) for the from-scratch training experiments.
- **Datasets**: For supervised fine-tuning experiments, we use 500 samples from CIFAR10 (Krizhevsky, 2009), a widely used benchmark for image classification tasks (De et al., 2022; Tobaben et al., 2023) and 5K samples from SST-2 (Socher et al., 2013) for text classification task. To train models from scratch, we use 50K samples from Purchase100 (Shokri et al., 2017).
- **Privacy Accounting**: We adapt Microsoft's `prv-accountant` (Gopi et al., 2021) to compute the theoretical upper bounds for substitute adjacency-based DP with Poisson subsampling. We share the code for this accountant in supplementary materials.
- **Hyperparameters**: We tune the noise added for DP relative to the subsampling rate $q$ and training steps $T$. We keep the other training hyperparameters fixed to isolate the effect of privacy amplification by subsampling (Bassily et al., 2014; Balle et al., 2018) on auditing performance. Detailed description of the hyperparameters used in our experiments is provided in Table A1.
- **Auditing Privacy Leakage / Step**: We perform step-wise audits by treating the model at each training step $t$ as a provisional model released to the adversary. The adversary is restricted to use only current model's parameters or outputs to compute the empirical privacy leakage at step $t$.

**Computing Empirical $\varepsilon$ with Gaussian DP (Dong et al., 2019):** DP (by Definition 1) implies an upper bound on the adversary's capability to distinguish between $\mathcal{M}(\mathcal{D})$ and $\mathcal{M}(\mathcal{D}')$. For computing the corresponding empirical lower bound on $\varepsilon$, we use the method prescribed by Nasr et al. (2023) which relies on $\mu$-GDP. This method allows us to get a high confidence estimate of $\varepsilon$ with reasonable repeats of the training algorithm.

Given a set of observations $\mathcal{O}$ and corresponding ground truth labels $\mathcal{B}$ obtained from Algorithm 1, the auditor can compute the False Negatives (FN), False Positives (FP), True Negatives (TN), and True Positives (TP) at a fixed threshold. Using these measures, the auditor estimates upper bounds on the false positive rate ($\overline{\text{FPR}}$) and false negative rate ($\overline{\text{FNR}}$) by using the Clopper–Pearson method (Clopper & Pearson, 1934) with significance level $\alpha = 0.05$.

Kairouz et al. (2015) express privacy region of a DP algorithm in terms of FPR and FNR. DP bounds the FPR and FNR attainable by any adversary. Nasr et al. (2023) note that the privacy region for DP-SGD can be characterized by $\mu$–GDP (Dong et al., 2019). Thus, the auditor can use $\overline{\text{FPR}}$ and $\overline{\text{FNR}}$ to compute the corresponding empirical lower bound on $\mu$ in $\mu$-GDP,

$$\mu_{\text{lower}} = \Phi^{-1}(1 - \overline{\text{FPR}}) - \Phi^{-1}(\overline{\text{FNR}}), \tag{5}$$

where $\Phi$ represents the cumulative density function of standard normal distribution $\mathcal{N}(0, 1)$. This lower bound on $\mu$ can be translated into a lower bound on $\varepsilon$ given a $\delta$ in $(\varepsilon, \delta)$-DP using the following theorem,

**Theorem 5.1** (Dong et al. (2019) Conversion from $\mu$-GDP to $(\varepsilon, \delta)$-DP). *If an algorithm $\mathcal{M}$ is $\mu$-GDP, then it is also $(\varepsilon, \delta)$-DP ($\varepsilon \geq 0$), where*

$$\delta(\varepsilon) = \Phi\left(-\frac{\varepsilon}{\mu} + \frac{\mu}{2}\right) - e^{\varepsilon}\Phi\left(-\frac{\varepsilon}{\mu} - \frac{\mu}{2}\right). \tag{6}$$

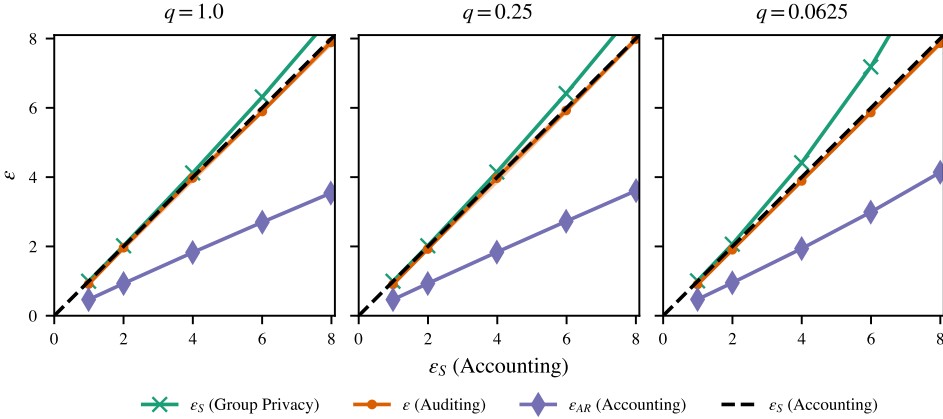

Figure 1: **Auditing DP using worst-case dataset canaries based on substitute adjacency.** When the adversary crafts the neighbouring datasets as worst-case dataset canaries (S1), we find that the empirical privacy leakage for a DP algorithm, $\varepsilon$ (Auditing ), exceeds the privacy upper bound for add/remove DP, $\varepsilon_{AR}$ (Accounting). It closely tracks the privacy budget predicted by substitute accountant, $\varepsilon_S$ (Accounting). The plot shows that $\varepsilon_S$ (Accounting) is tighter when compared to that $\varepsilon_S$ (Group Privacy) computed using Theorem 4.1. We fix $\delta_{\text{target}} = 10^{-5}$, $C = 1.0$ and $T = 500$. The auditing estimates are averaged over 3 repeats. For each repeat, we use $R = 25\text{K}$ runs to estimate $\varepsilon$ (Auditing) at the final step of training. The error bars represent $\pm 2$ standard errors around the mean computed over 3 repeats of auditing algorithm.

# 6 RESULTS

## 6.1 AUDITING WITH WORST-CASE CRAFTED DATASET CANARIES

Figure 1 depicts the relation between $\varepsilon_S$ (Accounting) computed with a substitute accountant, $\varepsilon_S$ (Group Privacy) computed using Theorem 4.1, $\varepsilon$ (Auditing) using crafted worst-case dataset canaries from Section 3.1, and $\varepsilon_{AR}$ (Accounting) computed with an add/remove accountant for a set

of DP parameters. We observe that $\varepsilon$ (Auditing) exceeds $\varepsilon_{AR}$ (Accounting) but remains tight with respect to $\varepsilon_S$ (Accounting). Thus, mounting a substitute-style attack using worst-case dataset canaries enables the adversary to detect whether $\mathcal{D}$ or $\mathcal{D}'$ was used for training a model with higher confidence than promised by $\varepsilon_{AR}$ (Accounting).

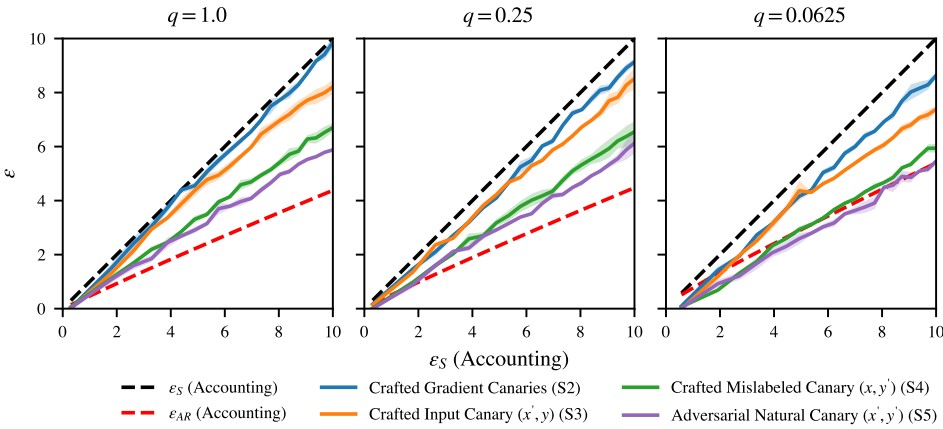

Figure 2: **Auditing models trained with DP using natural datasets.** We fine-tune final layer of ViT-B-16 models pretrained on ImageNet21K using CIFAR10. The privacy leakage ($\varepsilon$) audited using our proposed canaries for this setting exceeds the add/remove DP upper bounds, $\varepsilon_{AR}$ (Accounting). As these canaries are used to mount a substitute-style attack, the figure shows that add/remove DP overestimates protection against such attacks. Efficacy of the canaries decline as subsampling rate $q$ decreases, the effect being most significant for audits using input-space canaries. We plot $\varepsilon$ for every $k$th step ($k = 25$) of training averaged over 3 repeats of the auditing algorithm. For each repeat, we train $R = 2500$ models, $1/2$ trained with $z$ and the remaining with $z'$. The error bars represent $\pm 2$ standard errors around the mean computed over 3 repeats of auditing algorithm.

## 6.2 Auditing Models Trained with Natural Datasets

In this section, we report auditing results on models trained with natural datasets. In fine-tuning experiments with CIFAR10, all are proposed canaries outperform add/remove DP at large subsampling rates. With the strongest canaries, we observe that the empirical privacy leakage exceeds the add/remove DP upper bounds for models trained from scratch with Purchase100. Our proposed canaries have no discernible effect on the utility of the models as shown in Figure A1.

### 6.2.1 Using Gradient-Space Canaries

Figure 2 shows that, when auditing models that are trained using natural datasets, we get the tightest estimates of $\varepsilon$ by using crafted gradient canaries for auditing. The empirical privacy leakage ($\varepsilon$) estimated using these canaries violates $\varepsilon_{AR}$ (Accounting). The canary gradients, $g_z$ and $g_{z'}$, crafted using Algorithm 2 stay constant over the course of training and have near-saturation gradient norms ($\|g_z\| = \|g_{z'}\| = C$). This ensures that their effect on the parameter updates of the model is consistent and is most affected by the choice of subsampling rate $q$. As $q$ decreases, the canary is less visible to the model during training, which yields weaker audits.

### 6.2.2 Using Input-Space Canaries

In this setting, the adversary is only permitted to insert a crafted input record into the training dataset. In Figure 2, we observe that although input-space canaries yield less tight audits when compared to crafted gradient canaries, the privacy leakage audited using the input-space canaries can exceed the guarantees of add/remove DP. We observe that the efficacy of audits with input-space canaries decreases for later training steps. This deterioration is much more significant at a low subsampling rate ($q$). Additionally, in Appendix A.2, we observe that audits using input-space canaries are sensitive to the choice of other training hyperparameters such as clipping bound $C$ (Figure A2), number of training steps $T$ (Figure A3), and learning rate $\eta$ (Figure A4).

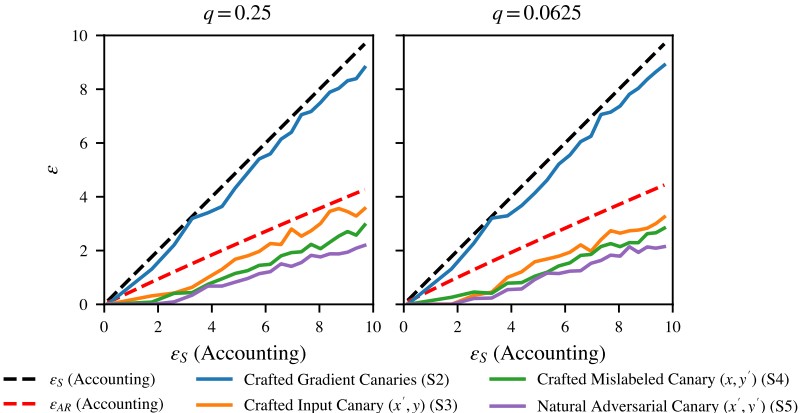

Figure 3: **Auditing MLP model trained from scratch with random initialization using Purchase100.** We find that auditing such models using input-space canaries yield weaker audits. We do not observe $\varepsilon$ from such audits to exceed the privacy implied by $\varepsilon_{AR}$ (Accounting). However, using crafted gradient canaries, we still get $\varepsilon$ from auditing which is consistent with $\varepsilon_S$ (Accounting). We plot $\varepsilon$ for every $k$th step ($k = 125$) of training. We train $R = 2500$ models, $1/2$ trained with $z$ and the remaining with $z'$. We use DP-Adam as the optimizer for training models from scratch.

### 6.2.3 AUDITING MODELS TRAINED FROM SCRATCH

Training models from scratch with random initialization is a non-convex optimization problem. Figure 3 shows that auditing models trained from scratch on Purchase100 dataset using input-space canaries yields weaker audits. We find that input-space canaries are sensitive to model initialization and the choice of optimizer (DP-Adam in this case). Subsampling further deteriorates the effectiveness of audits with input-space canaries. In this setting, add/remove DP does suffice to protect against attacks using input-space canaries as shown in Figure 3. However, our proposed crafted gradient canaries still yield strong audits for models trained from scratch with empirical privacy leakage that closely follows $\varepsilon_S$ (Accounting).

### 6.3 AUDITING MODELS FINE-TUNED FOR TEXT CLASSIFICATION

We fine-tune a linear layer on top of Sentence-BERT (Reimers & Gurevych, 2019) encoder using 5K samples from Stanford's Sentiment Treebank (SST-2) dataset (Socher et al., 2013). We present the results for this experiment in Figure A6. The models are trained using DP-SGD. We find that gradient-canary-based auditing yields tight results. While the audits using input-space canaries are not tight, we do observe that the empirical privacy leakage estimated using them does exceed the privacy guaranteed by add/remove DP.

## 7 DISCUSSION AND CONCLUSION

We provide empirical evidence which shows that for certain ML models, DP with add/remove adjacency will not offer adequate protection against attacks such as attribute inference at the level guaranteed by the privacy parameters. This is because the threat model for these attacks mimics substitute-style attacks. In Figure 2, for DP models are trained using natural datasets, we observe violations of add/remove DP guarantees with the canaries designed to substitute a target record or a target record's gradient in the training dataset. The resulting empirical privacy leakage from such audits closely follows DP upper bound for substitute adjacency. Thus, practitioners seeking attribute or label privacy using standard DP libraries which default to add/remove adjacency-based accountants might risk overestimating the protection add/remove DP affords against substitute-style attacks.

We observe that fine-tuned models (as shown in Figure 2) are more prone to privacy leakage with input-space canaries compared to models trained from scratch (Figure 3). In practice, limited sensitive data makes DP training from scratch challenging. Tramèr & Boneh (2021) have shown that given a suitable public pretraining dataset, fine-tuning a pretrained model on sensitive data can yield higher utility than models trained from scratch. This makes our results with supervised

fine-tuning important since it reveals that poisoning the fine-tuning datasets once with input-space canaries is sufficient to cause privacy leakage exceeding add/remove DP bounds, particularly at large subsampling rates which are often used for improved privacy–utility trade-off (De et al., 2022; Mehta et al., 2023).

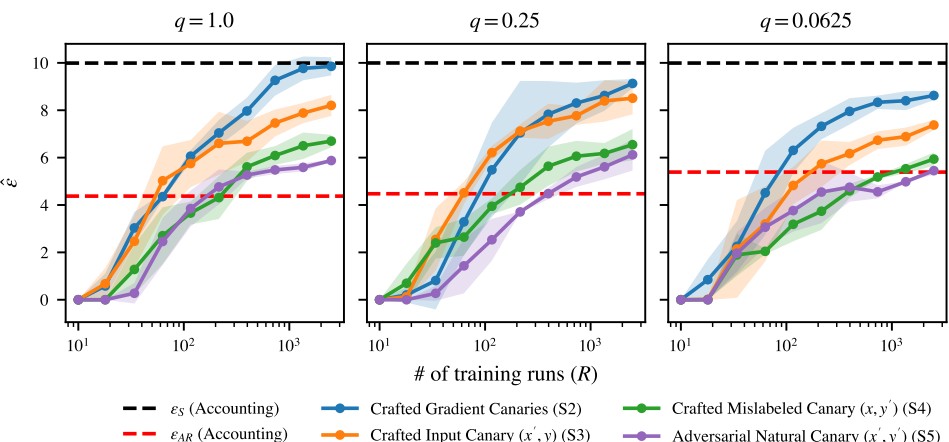

Figure 4: **Effect of number of training runs $R$ on privacy auditing.** For ViT-B-16 models with final layer fine-tuned on CIFAR10 ($T = 500, C = 2.0$), we record the effect of change in $R$ on the empirical privacy leakage $\hat{\varepsilon}$, at the final step of training. The error bars represent $\pm 2$ standard errors around the mean computed over 3 repeats of auditing algorithm. In each repeat, $1/2$ of the models are trained with $z$ and the remaining with $z'$.

Our methods to audit DP under substitute adjacency are not without limitations. We note that the efficacy of our proposed input-space canaries depends strongly on the training hyperparameters (see Figures A2 to A4 in Appendix A.2). They provide weaker audits at later training steps, especially when the training problem involves non-convex optimization and a low subsampling rate $q$. This has been a persistent issue with input-space canaries as noted by Nasr et al. (2023). Our results show that canaries with consistent gradient signals and near-saturation gradient norms are most robust to the effect of training hyperparameters. An interesting direction for future work is to design input-space canaries that are robust to training hyperparameters and yield tight audits for models trained with real, non-convex objectives.

Our canaries are tailored to audit gradient-based DP algorithms, such as DP-SGD. We expect the canaries to work well with other gradient-based methods, such as DP-Adam, although some performance degradation is possible (as seen in Figure 3). We do not expect our proposed auditing approach to extend to other DP mechanisms which operate differently. For instance, label DP (Chaudhuri & Hsu, 2011) is a special case of substitute DP, where you only substitute the label of an example. Auditing using a crafted mislabeled canary is the same threat model as label DP. As substitute DP is a generalization of label DP, it will also be valid for auditing a substitute DP mechanism, even though it might not be optimal for that. While DP-SGD with substitute accounting is a valid label DP mechanism, in practice, label DP is implemented using very different methods (Ghazi et al., 2021; 2024; Busa-Fekete et al., 2023; Zhao et al., 2025). As such, our auditing techniques would not be suitable for those methods.

Furthermore, our methods for privacy auditing rely on multiple repeats of the training process to obtain a high confidence measure of lower bound on $\varepsilon$. In Figure 4, we observe that with limited number of runs, there is a risk of underestimating the privacy leakage. At low subsampling rate ($q$), the continuous upward trend of auditing curves show that the process has not converged, even with $R = 2500$ runs. For a detailed breakdown of the computational cost of the our method, we refer to Table A2. While our method is computationally expensive, it could potentially be optimized by integrating single-run auditing approaches (Steinke et al., 2023; Mahloujifar et al., 2025), although this might involve a trade-off between computational efficiency and the strength of the resulting audits.

## ACKNOWLEDGMENTS

This work was supported by the Research Council of Finland (Flagship programme: Finnish Center for Artificial Intelligence, FCAI, Grant 356499 and Grant 359111), the Strategic Research Council at the Research Council of Finland (Grant 358247) as well as the European Union (Project 101070617). Views and opinions expressed are however those of the author(s) only and do not necessarily reflect those of the European Union or the European Commission. Neither the European Union nor the granting authority can be held responsible for them. This work has been performed using resources provided by the CSC– IT Center for Science, Finland (Project 2003275). The authors acknowledge the research environment provided by ELLIS Institute Finland. We would like to thank Ossi Räisä and Marlon Tobaben for their helpful comments and suggestions.

## REPRODUCIBILITY STATEMENT

The code for our experiments is available at: `https://github.com/DPBayes/limitations_of_add_remove_adjacency_in_dp`. We adapted the code from Tobaben et al. (2023) for the fine-tuning experiments.

## ETHICS STATEMENT

The research conducted in the paper conform, in every respect, with the ICLR Code of Ethics (`https://iclr.cc/public/CodeOfEthics`).

## USE OF LARGE LANGUAGE MODELS (LLMS)

We used LLMs to polish the content of this manuscript for readability and conciseness. We also used it to improve the presentation of mathematical content with LaTeX. LLMS were not used to generate any novel content.

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

## A  APPENDIX

### A.1  EXPERIMENTAL TRAINING DETAILS

Table A1 details the hyperparameters used for training the models for our experiments. We use Opacus (Yousefpour et al., 2021) to facilitate DP training of models with Pytorch (Paszke et al., 2019).In our experiments, we vary the seed per run, which ensures randomness in mini-batch sampling and, in the case of models trained from scratch, also ensures random initialization per run.

We find that adding a canary to the gradients or datasets does not compromise the utility of the trained models which we measure in terms of their accuracy on the test dataset. Figure A1 compares the test accuracies for models poisoned using gradient canaries (Algorithm 2) and crafted input canary (Algorithm 3) to models trained with the target record. With $q = 1$, the model "sees" the canary at each step of training. Despite that, we observe minimal difference in test accuracies averaged across 5 models trained with target record and models trained with either gradient or crafted input canaries.

### A.2  EFFECT OF TRAINING HYPERPARAMETERS ON AUDITING

Choice of the clipping bound $C$ only affects audits done using input-space canaries significantly. This is because gradient-space canaries are crafted using Algorithm 2 which ensures that $\|g_z\|$ and $\|g_{z'}\| = C$ (that is, they have near-saturation gradient norms) throughout the training process.

Table A1: Hyperparameters used for the experiments in the main paper. We use these as default hyperparameters for a given dataset unless otherwise specified.

| Hyperparameters | CIFAR10 | Purchase100 | SST-2 |
|---|---|---|---|
| DP Optimizer | DP-SGD | DP-Adam | DP-SGD |
| Trainable Parameter Count ($|\theta|$) | 768 | 89828 | 384 |
| Initialization ($\theta_0$) | Fixed | Random | Fixed |
| Subsampling Rate ($q$) | $(1.0, 0.25, 0.0625)$ | $(0.25, 0.0625)$ | $(1.0, 0.25)$ |
| Clipping Bound ($C$) | 2.0 | 5.0 | 2.0 |
| Training Steps ($T$) | 500 | 2500 | 2500 |
| Learning Rate $\eta$ | 0.001 | 0.0018 | 0.01 |
| | **Common Settings** | | |
| Loss Function | Cross Entropy Loss | | |
| Subsampling | Poisson | | |
| Auditing Runs ($R$) | 2500 | | |
| $\delta$ | $10^{-5}$ | | |

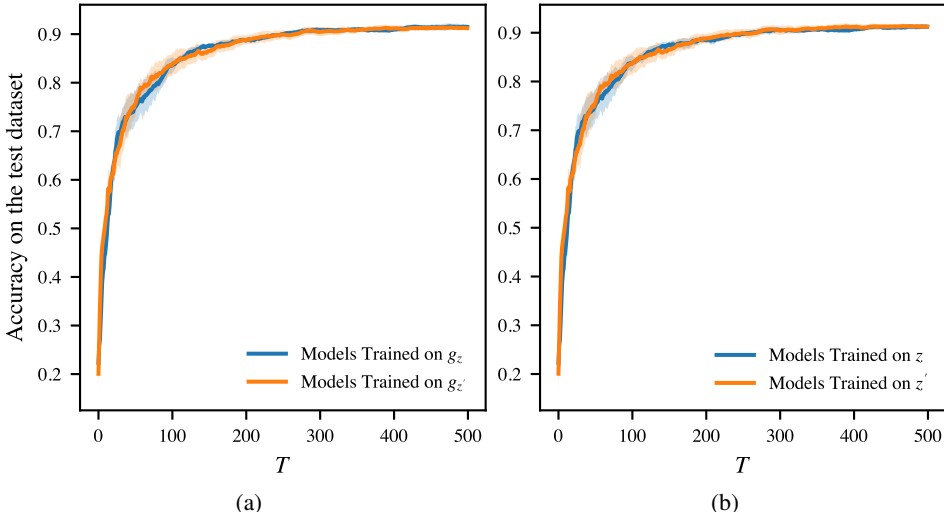

(a)                                                      (b)

Figure A1: **Auditing with our proposed canaries does not compromise model utility.** The figure depicts test accuracies as observed over the course of training for (a) models trained with gradient canaries (Algorithm 2), and (b) models trained on crafted input canary (Algorithm 3). The model is ViT-B-16 pretrained on ImageNet21K with final layer fine-tuned on CIFAR10. We train the model with $q = 1$ for 500 steps with $\varepsilon = 10, \delta = 10^{-5}$ for substitute DP.

Thus, the crafted gradient canaries are minimally affected by clipping during training. In contrast, input-space canaries, specifically, crafted input (Algorithm 3) and adversarial natural canaries (Algorithm 5) show high sensitivity to the choice of $C$. High $C$ corresponds to higher noise added during DP which affects the distinguishability between target sample and the canary.

In Figure A3, we find that, keeping subsampling rate $q$ fixed ($= 0.0625$), if we vary the number of training steps $T$, it affects the auditing with input-space canaries. For a fixed $q$, a larger $T$ means that the canary is "seen" more number of times during training. As we keep the total privacy budget constant, a larger $T$ for a fixed $q$ also implies an increase in the noise accumulated over intermediate steps. We observe that the audits done with crafted input canary and adversarial natural canaries suffer with an increase in $T$, especially at later training steps.

Similarly, Figure A4 demonstrates that auditing done with input space canaries is affected by the choice of learning rate. Thus, we find that canaries crafted/ chosen to mimic samples from training data are susceptible to the training hyperparameters. In auditing, we assume that the adversary has access to the hyperparameters. However, in practice, the model trainer might choose to keep these

hyperparameters confidential. This means that the audits done using such canaries can underestimate privacy leakage suggested by formal DP guarantees.

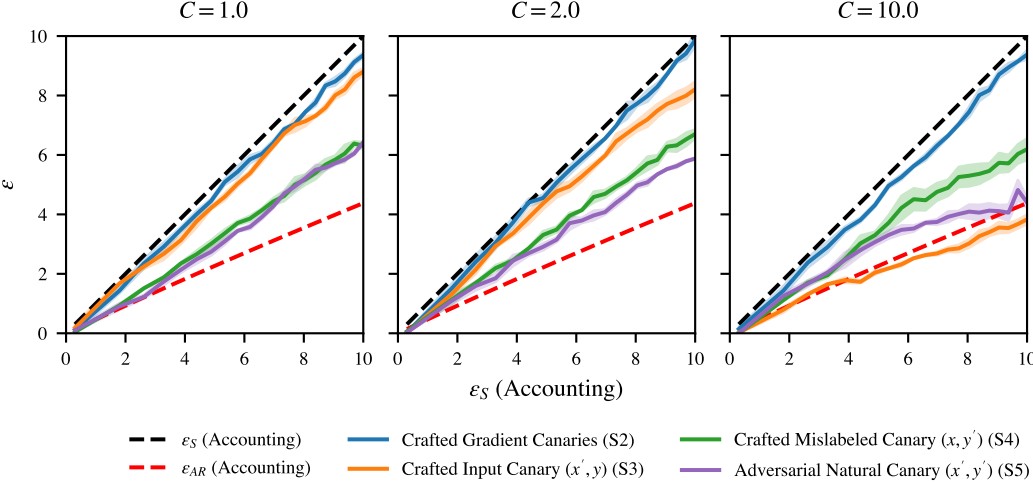

Figure A2: **Effect of clipping bound $C$ on privacy auditing.** For ViT-B-16 models with final layer fine-tuned on CIFAR10 (with $q = 1.0, T = 500$), varying $C$ causes crafted input and adversarial natural canary to loose their effectiveness as $C$ increases. Higher $C$ leads to higher per-step noise added during training. This adversely affects the audits using crafted input and adversarial natural canary. Crafted gradient and crafted mislabeled canary show relatively less sensitivity to $C$. We plot $\varepsilon$ for every $k$th step ($k = 25$) averaged over 3 repeats of the auditing algorithm. For each repeat, we train $R = 2500$ models, $1/2$ trained with $z$ and the remaining with $z'$. The error bars represent $\pm 2$ standard errors around the mean computed over 3 repeats of auditing algorithm.

## A.3 Relationship Between Expected Privacy Loss Under Substitute DP And Add/Remove DP

Typically, the privacy loss under substitute DP is expected to be $2\times$ the privacy loss under add/remove DP. However, as shown in Equation (4), this holds true when the $\delta$ is also scaled appropriately when moving from add/remove to substitute DP. If we keep the $\delta$ constant for add/remove and substitute DP, $\varepsilon_{SR}$ can be $> 2\varepsilon_{AR}$, especially when $\varepsilon$ is large, that is, when we use a large subsampling rate ($q$) and low noise ($\sigma$), as shown in Figure A5. We also show that this ratio is dependent on changes in $q$ and $\sigma$.

## A.4 Additional Results / Tables

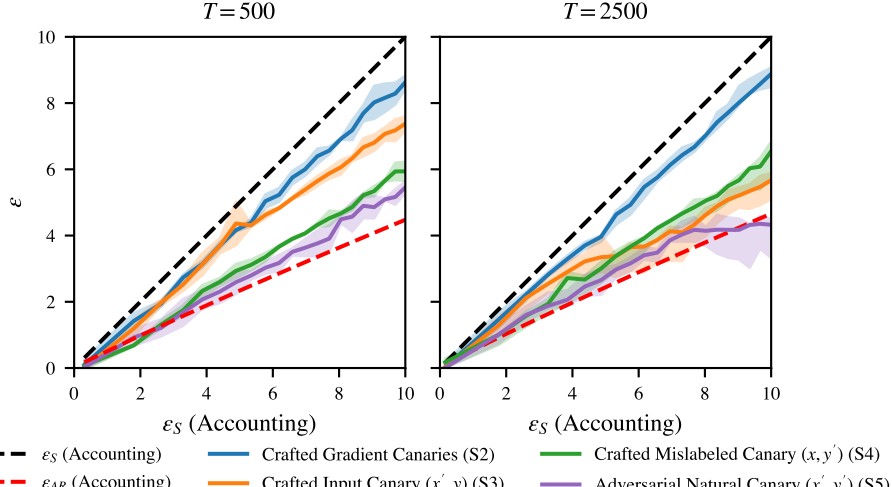

Figure A3: **Effect of training steps $T$ on privacy auditing.** For ViT-B-16 models with final layer fine-tuned on CIFAR10 (with $q = 0.0625, C = 2.0$), varying $T$ with subsampling leads to an increase in the noise accumulated over intermediate steps between successive canary appearances during training. This most significantly affects auditing with crafted input and adversarial natural canary. They yield relatively stronger audits for $T = 500$ but with $T = 2500$, they loose their efficacy for later training steps. As the total privacy budget is fixed for $T = 500$ and $T = 2500$, the degradation in audits for input-space canaries can be attributed to the higher per-step noise associated with larger $T$. We plot $\varepsilon$ for every $k$th step ($k = 25$ for $T = 500$ and $k = 125$ for $T = 2500$) averaged over 3 repeats of the auditing algorithm. For each repeat, we train $R = 2500$ models, $1/2$ trained with $z$ and the remaining with $z'$. The error bars represent $\pm 2$ standard errors around the mean computed over 3 repeats of auditing algorithm.

Table A2: Computational cost breakdown for different phases of the auditing schema (Algorithm 1).

| **Phase I: Crafting Canaries for Auditing** | **Computational Cost** |
|---|---|
| *Common cost for all canary types* | |
| Training the reference model | $\Omega(T \times P_{\text{train}})$ |
| *Additional cost (incurred only if the corresponding canary is crafted)* | |
| Crafting Gradient Canary (Algorithm 2) | $+ \Theta(P_{\text{train}})$ |
| Crafting Input Canary (Algorithm 3) | $+ \Theta(N \times P_{\text{train}})$ |
| Crafting Mislabeled Canary (Algorithm 4) | $+ \Theta(|\mathcal{Y}| \times P_{\text{train}})$ |
| Crafting Adversarial Natural Canary (Algorithm 5) | $+ \Theta(|\mathcal{D}_{\text{aux}}| \times P_{\text{train}})$ |
| **Phase II: Training Multiple Instances of Target Model** | |
| Training $R$ instances of the target model | $+ \Omega(R \times T \times P_{\text{train}})$ |
| **Phase III: Computing Empirical $\varepsilon$** | |
| Post-processing an $R \times T$ array of distinguishability scores | $+ \Omega(R \times T)$ |

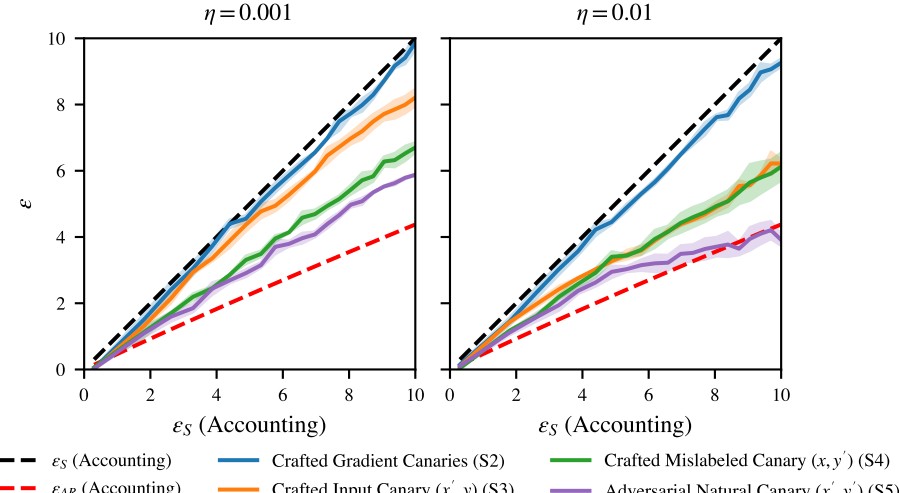

Figure A4: **Effect of learning rate $\eta$ on privacy auditing.** For ViT-B-16 models with final layer fine-tuned on CIFAR10 (with $q = 1.0, T = 500$), change in $\eta$ reduces the effectiveness of audits with input-space canaries. We plot $\varepsilon$ for every $k$th step ($k = 25$) averaged over 3 repeats of the auditing algorithm. For each repeat, we train $R = 2500$ models, $1/2$ trained with $z$ and the remaining with $z'$. The error bars represent $\pm 2$ standard errors around the mean computed over 3 repeats of auditing algorithm.

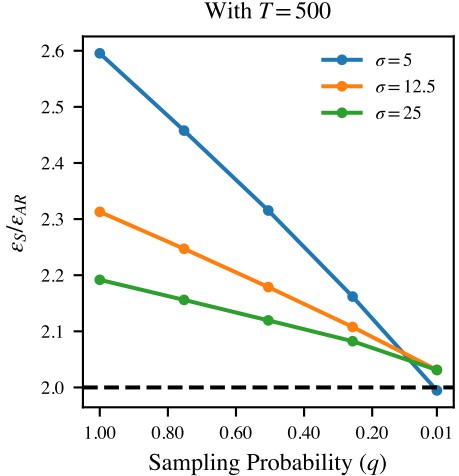

Figure A5: **Relationship between $\varepsilon_S$ (Accounting) and $\varepsilon_{AR}$ (Accounting) for varying Subsampling Rate ($q$) and Noise ($\sigma$).** The relationship between $\varepsilon_S$ and $\varepsilon_{AR}$ as defined by Equation (4) is expected to hold when $\delta_S = (1 + e^{\varepsilon_{AR}})\delta_{AR}$. However, for a fixed $\delta_S = \delta_{AR} = 10^{-5}$, we find that $\varepsilon_S$ can be $> 2\varepsilon_{AR}$, especially for large $q$ and low $\sigma$.

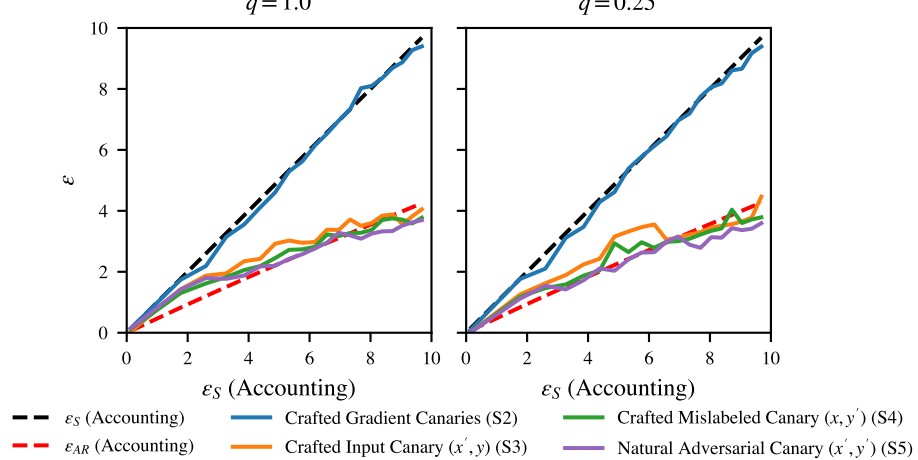

Figure A6: **Auditing Models Trained For Text Classification.** We audit Sentence-BERT models with final linear layer fine-tuned on SST-2 dataset ($C = 2.0$, $T = 2500$). We find that using our canaries, we can extract privacy leakage from these models which may exceed the privacy guaranteed by add/remove DP but is in line with the guarantees of substitute DP. We plot $\varepsilon$ for every $k$th step ($k = 125$) of training. For each repeat, we train $R = 2500$ models, $1/2$ trained with $z$ and the remaining with $z'$.

