# OpenReview forum: "Beyond Membership: Limitations of Add/Remove Adjacency in Differential Privacy"
_ICLR.cc/2026/Conference — ICLR 2026 Poster_

### Official Review · Reviewer_tuq2 · 2025-10-30

**Soundness:** 3
**Presentation:** 3
**Contribution:** 3
**Rating:** 6
**Confidence:** 5

**Summary:**

This paper identifies an important gap between the privacy guarantees provided by add/remove adjacency and the actual protection needed for attribute inference in differential privacy. The authors develop novel auditing techniques for substitute adjacency and demonstrate empirically that standard add/remove accounting can significantly overstate attribute privacy protection.

**Strengths:**

1. The paper addresses a critical misconception in practical DP deployments where practitioners may unknowingly overestimate protection against attribute inference when using standard libraries.
2. The paper explains with examples and citations (Sections 1–2, pp. 1–2), why add/remove adjacency may fail for attribute privacy. This makes the motivation persuasive.
3. Experiments in Section 6 demonstrate that empirical ε values from auditing exceed add/remove accountant bounds. This supports the paper’s key claim with direct quantitative evidence.
4. Reproducibility is good since a public anonymized code repository is provided, and Appendix Table A1 includes detailed hyperparameter.

**Weaknesses:**

1. The attacks are targeting DP-SGD; Whether they are applicable to other mechanisms (e.g., DP AdamW, DP sampling-based methods) are not clear.
2. Although empirical ε values are averaged over multiple runs (Figure 1, p. 7), the paper does not report confidence intervals or variance across repetitions, making it hard to assess statistical robustness.
3. The auditing method requires training thousands of models (e.g., R = 2500 per setting). This raises concerns about scalability and applicability to larger models or datasets.

**Questions:**

1. Whether this can be generalized to other mechanisms rather than just DP-SGD?
2. How robust are the empirical ε estimates to randomness in mini-batch sampling and initialisation?

---

> ### Author Response · Authors · 2025-11-25
> **Official Comment By Authors**
>
> We thank the reviewer for their time to review our manuscript and for their valuable feedback. We address the weaknesses/questions put forth by them below:
>
> > Whether this can be generalized to other mechanisms rather than just DP-SGD?
>
> Our canaries are tailored to audit gradient-based DP algorithms, such as DP-SGD. We expect the canaries to work well with other gradient-based methods, such as DP-Adam,  although some performance degradation is possible (as seen in Figure 3). We do not expect the approach to extend to DP mechanisms that operate differently. We make a note of this in the revised Section 7 (Discussion and Conclusion).
>
> > How robust are the empirical ε estimates to randomness in mini-batch sampling and initialisation?
>
> In our experiments, we vary the seed per run, which ensures randomness in mini-batch sampling and, in the case of models trained from scratch, also ensures random initialisation per run. We have amended Section A1 (Experimental Training Details) to clarify this.
>
> > Although empirical ε values are averaged over multiple runs (Figure 1, p. 7), the paper does not report confidence intervals or variance across repetitions, making it hard to assess statistical robustness.
>
> In Figure 1 (Page 7), we do plot $\pm 2$ standard error around the mean for most figures over 3 repeats of the auditing algorithm, but, owing to the large number of runs used to estimate empirical $\varepsilon$ ($R=25$K), they are quite narrow. We have updated the caption of figures to report that the error bars represent $\pm 2$ standard errors around the mean, computed over the 3 repeats of the auditing algorithm.
>
> > The auditing method requires training thousands of models (e.g., R = 2500 per setting). This raises concerns about scalability and applicability to larger models or datasets.
>
> We have added Figure 4 in the paper, which illustrates the effect of runs ($R$) on the quality of the resulting audits. Using a low number of runs can lead to an underestimation of empirical privacy leakage during auditing. We have amended Section 7 (Discussion and Conclusion) to highlight this issue.

---

### Official Review · Reviewer_Hu7w · 2025-10-31

**Soundness:** 3
**Presentation:** 3
**Contribution:** 3
**Rating:** 6
**Confidence:** 4

**Summary:**

The paper argues that when the goal is to protect per-record attributes (e.g., labels) rather than just membership, reporting DP guarantees under the standard add/remove adjacency can be misleading. The authors propose auditing methods under substitute (replace-one) adjacency—including worst-case “canary” constructions in gradient space and input space—and show empirically that models trained with DP-SGD can leak more than the add/remove accountant would suggest, while the leakage aligns with a substitute-adjacency accountant.

**Strengths:**

The paper spotlights a real deployment pitfall: most libraries/accountants default to add/remove adjacency, yet many fine-tuning scenarios care about attribute privacy for users already known to be in the training data.

The gradient-space and input-space canaries (plus mislabeled and natural choices) are well-motivated and easy to reproduce. The “worst-case dataset canary” analysis gives intuitive mixture-of-Gaussians distinguishers and ties directly to the accountant.

**Weaknesses:**

The paper mixes hidden-state audits, crafted gradients, and input-space canaries; readers may struggle to map these to real-world attacker access. Suggestion: Add a single table that contrasts black-box / hidden-state / white-box access (what the adversary sees, what they can inject) and mark which canaries apply to each, plus a “practical examples” column (e.g., data-collection poisoning during SFT; known-in-training user seeking label privacy).

CIFAR-10 fine-tuning and Purchase100 MLP are reasonable, but the paper’s practical claim (“attribute privacy can be overstated”) would be stronger with at least one text or tabular SFT workload with label privacy relevance.

**Questions:**

I have no questions about this paper.

---

> ### Author Response · Authors · 2025-11-25
> **Official Comment By Authors**
>
> We thank the reviewer for their time to review our manuscript and for their valuable feedback. We address the weaknesses/questions put forth by them below:
>
> > The paper mixes hidden-state audits, crafted gradients, and input-space canaries; readers may struggle to map these to real-world attacker access. Suggestion: Add a single table that contrasts black-box / hidden-state / white-box access (what the adversary sees, what they can inject) and mark which canaries apply to each, plus a “practical examples” column (e.g., data-collection poisoning during SFT; known-in-training user seeking label privacy).
>
> We have modified Table 1 to include an additional column to clarify the threat model/ access associated with each auditing approach. We have added practical examples where our proposed canaries can be used in the main text (lines 228-230 in Section 3.2.1 and lines 238-239 in Section 3.2.2).
>
> > CIFAR-10 fine-tuning and Purchase100 MLP are reasonable, but the paper’s practical claim (“attribute privacy can be overstated”) would be stronger with at least one text or tabular SFT workload with label privacy relevance.
>
> Heeding the reviewer’s suggestion, we have added Section 6.3 (Page 9) in the paper, where we present the results of auditing the Sentence-BERT model with the final linear layer fine-tuned on Stanford’s Sentiment Treebank (SST-2) dataset.

---

### Official Review · Reviewer_GPRK · 2025-11-01

**Soundness:** 4
**Presentation:** 4
**Contribution:** 3
**Rating:** 8
**Confidence:** 4

**Summary:**

The paper shows that when we are interested in attribute DP, add/remove adjacency overstates privacy. Substitute adjacency, which allows replacing a record, better captures attribute privacy. Using canary-based attacks to audit DP under substitute adjacency, the authors find that the privacy leakage can exceed add/remove guarantees but aligns with substitute bounds, thus choice of adjacency is critical for per-record attribute protection in practical deployments.

**Strengths:**

1) The authors provides a clear and rigorous demonstration that standard add/remove adjacency overstates privacy when the goal is to protect individual labels. Through extensive experiments (Section 5), the authors quantify the gap between theoretical DP guarantees and the actual leakage observed in practice, showing that models can leak significantly more information about labels than add/remove accounting would suggest.

2) The main strength of the work is its development of a novel auditing framework under substitute adjacency. The authors design canary-based attacks that can replace target records to probe privacy leakage, with Algorithm 4 being effective in estimating tight lower bounds. By crafting canaries in both the input and gradient spaces, the study establishes a robust methodology that reliably captures attribute-level privacy loss.

**Weaknesses:**

Computational Cost: High-confidence audits require multiple retrainings, which can be resource-intensive. The authors acknowledge this and suggest potential optimizations using single-run approaches (Steinke et al., 2023).

(minor) Title: The title does not explicitly mention attribute or label privacy. Adding terms like “attribute DP” or “label DP” could make the paper’s setting immediately clear to readers.

**Questions:**

1. Given the high computational overhead, what is the authors' estimated potential reduction in the required number of training runs that could be achieved by integrating robust canaries with a single-run auditing technique?

2. "The mislabeled canary is relatively less sensitive to $C$, likely due to systematic misalignment with the task objective", can you elaborate on this?

---

> ### Author Response · Authors · 2025-11-25
> **Official Comment By Authors**
>
> We thank the reviewer for their time in reviewing our manuscript and for their valuable feedback. We address the weaknesses/questions put forth by them below:
>
> > Given the high computational overhead, what is the authors' estimated potential reduction in the required number of training runs that could be achieved by integrating robust canaries with a single-run auditing technique?
>
> We have added Figure 4, which illustrates the effect of the number of training runs on the efficacy of the audits. As mentioned in the paper, an interesting direction for future work would be to craft robust canaries that can be used in conjunction with auditing approaches using $O(1)$ training run, such as those proposed by Steinke et al., Mahloujifar et al. However, as noted in these works, this might come at the cost of weaker or highly variable audit estimates.
>
>
> > "The mislabeled canary is relatively less sensitive to
> , likely due to systematic misalignment with the task objective", can you elaborate on this?
>
> We have removed this statement from Section A2, as it is unclear what causes the observed differences in sensitivity to changes in $C$ between different crafting methods.

---

> > ### Comment · Reviewer_GPRK · 2025-11-28
> >
> > Thank you for the clarifications, I recommend that the paper be accepted

---

### Official Review · Reviewer_hf2R · 2025-11-02

**Soundness:** 3
**Presentation:** 2
**Contribution:** 3
**Rating:** 4
**Confidence:** 2

**Summary:**

This paper examines how current differential privacy accounting, based on add/remove adjacency, fails to reflect real privacy risks in machine learning. The authors argue that when an attacker already knows a record is in the dataset and wants to infer attributes such as labels, the correct notion is substitute adjacency. They design a set of empirical canary audits in gradient space and input space to measure empirical lower bounds of $\epsilon$. Experiments on image and tabular data show that substitute adjacency leads to about twice the reported privacy loss of standard accounting. The work demonstrates that existing DP practices overestimate protection against attribute inference.

**Strengths:**

- The paper makes a valuable empirical contribution by quantitatively showing how privacy guarantees differ under two adjacency definitions. This is an important clarification for the community and helps link theoretical concepts to practical implications in training differentially private models.

- The originality of the work lies primarily in the empirical auditing framework. The audited privacy loss values align closely with the theoretical upper bounds, supporting the validity of the approach.

- Experimental details and configurations are described clearly, making the study repeatable.

**Weaknesses:**

- The paper discusses attribute inference but does not evaluate or contrast its findings with other DP formulations that also target label or feature privacy, for example, label-DP.

- It remains unclear what information the attacker is assumed to know. The experiments appear to assume full feature knowledge except for the sensitive attribute, but this is not formally stated or varied.

- Each privacy point requires many repeated training runs. The feasibility of scaling such auditing to larger settings remains unclear.

**Questions:**

- Would the tight bound derived under substitute adjacency remain valid when compared with mechanisms built on Label Differential Privacy, which is specifically designed to protect labels?
- Could the authors more precisely define the attacker’s prior knowledge in each auditing setup?
- Could the authors analyze the computational cost of their auditing approach and provide empirical approximations or results demonstrating its scalability to larger applications?
- How does the observed privacy loss behave when only part of the input features are assumed known to the adversary?
- For a fixed noise parameter, is there a typical ratio between substitute and add/remove privacy losses?

---

> ### Author Response · Authors · 2025-11-25
> **Official Comment by Authors**
>
> We thank the reviewer for their time in reviewing our manuscript and for their valuable feedback. We address the weaknesses/questions put forth by them below:
>
> > Would the tight bound derived under substitute adjacency remain valid when compared with mechanisms built on Label Differential Privacy, which is specifically designed to protect labels?
>
> Tight accounting is mechanism-specific. Label DP mechanisms are usually very different from regular DP mechanisms, and therefore require dedicated accounting.
>
> > Could the authors more precisely define the attacker’s prior knowledge in each auditing setup?
>
> We have added Table 2 in the paper, which details the adversary’s prior knowledge in each auditing setup.
>
> > Could the authors analyze the computational cost of their auditing approach and provide empirical approximations or results demonstrating its scalability to larger applications?
>
> As mentioned in the common response to all the reviewers, we have added Figure 4 (Page 10), which illustrates the effect of the number of training runs on the efficacy of the audits. We refer to the comment for details.
>
> > How does the observed privacy loss behave when only part of the input features are assumed known to the adversary?
>
> When crafting input-space canaries in our paper, we assume that the adversary has access to the target sample $z$ and uses this knowledge to craft an optimal canary for auditing. An adversary equipped with only partial input features might struggle to construct a strong canary if the known features do not substantially influence the model’s behaviour during training.
>
> > For a fixed noise parameter, is there a typical ratio between substitute and add/remove privacy losses?
>
> Typically, the expected privacy loss under substitute DP is twice the loss expected under add/remove DP due to the need to protect against the removal and addition of a sample from the target dataset. However, as shown in Equation 4 (Page 6) in the paper, this is true when $\delta_S = (1+e^{\varepsilon_{AR}})\delta_{AR}$. In our experiments, we keep the $\delta$ fixed for substitute and add/remove DP. Hence, we observe that for a fixed noise parameter, the ratio of $\varepsilon_S: \varepsilon_{AR}$ can be $>2$, especially when $\varepsilon$ is large, i.e. when using a large subsampling rate ($q$) and low noise $(\sigma)$. We refer to Figure A6 (Page 18) in the Appendix for more details on this.

---

> > ### Comment · Reviewer_hf2R · 2025-11-27
> >
> > I appreciate the authors' response and clarification. Could you please add a more detailed discussion part in the draft that clarifies the difference between substitute label adjacency DP and label DP?

---

> > > ### Author Response · Authors · 2025-11-27
> > > **Official Comment By Authors**
> > >
> > > > I appreciate the authors' response and clarification. Could you please add a more detailed discussion part in the draft that clarifies the difference between substitute label adjacency DP and label DP?
> > >
> > > Per the reviewer's suggestion, we have added the following paragraph detailing the difference between substitute DP and label DP in Section 7: Discussion and Conclusion (see lines 525-532) in the paper.
> > >
> > > *For instance, label DP is a special case of substitute DP, where you only substitute the label of an example. Auditing using a crafted mislabeled canary is the same threat model as label DP. As substitute DP is a generalization of label DP, it will also be valid for auditing a substitute DP mechanism, even though it might not be optimal for that. While DP-SGD with substitute accounting is a valid label DP mechanism, in practice, label DP is implemented using very different methods. As such, our auditing techniques would not be suitable for those methods.*

---

### Author Response · Authors · 2025-11-25
**Official Comment by Authors**

We thank all the reviewers for their time in reviewing our manuscript and for their valuable feedback. Below, we summarise the changes made to the paper to address the concerns raised by the reviewers:

- **Defining the attacker’s prior knowledge and access in each auditing setup**: We have updated Table 1 (Page 4) in the paper to include an additional column to clarify the threat model/ access associated with each auditing approach. Furthermore, we have added Table 2 (Page 4), which details the adversary’s prior knowledge in each auditing setup.

- **The computational cost of our proposed auditing approach**: We have added Figure 4 (Page 10) in the paper, which illustrates the effect of the number of training runs on the efficacy of the audits. Using only a small number of runs to reduce computational cost can cause the empirical privacy leakage of the DP mechanism to be underestimated. We have amended Section 7 (Discussion and Conclusion) to highlight this issue. As mentioned in the paper, an interesting direction for future work would be to craft robust canaries that can be used in conjunction with auditing approaches using O(1) training run, such as those proposed by Steinke et al.[1] , Mahloujifar et al. [2]. However, as noted in these works, this might come at the cost of weaker or highly variable audit estimates.

- **Auditing Models Trained For Text Classification Task**: We have added Section 6.3 (Page 9) in the paper, where we present the results of auditing the Sentence-BERT model with the final linear layer fine-tuned on Stanford’s Sentiment Treebank (SST-2) dataset.

- **Improved Figure Captions**: We have updated the caption of figures to report that the error bars represent $\pm 2$ standard errors around the mean, computed over the 3 repeats of the auditing algorithm.


[1] Thomas Steinke, Milad Nasr, and Matthew Jagielski. Privacy Auditing with One (1) Training Run. NeurIPS 2023.

[2] Saeed Mahloujifar, Luca Melis, and Kamalika Chaudhuri. Auditing $f$-Differential Privacy in One Run. ICML 2025.

---

> ### Author Response · Authors · 2025-12-01
> **Extended Official Comment By Authors**
>
> - **Clarifying the privacy mechanisms which can be audited using the proposed canaries**: Our canaries are tailored to audit gradient-based DP algorithms, such as DP-SGD. We expect the canaries to work well with other gradient-based methods, such as DP-Adam,  although some performance degradation is possible (as seen in Figure 3). We do not expect the approach to extend to DP mechanisms that operate differently. For instance, Label DP [3] is a special case of substitute DP, where you only substitute the label of an example. Auditing using a crafted mislabeled canary is the same threat model as label DP. As substitute DP is a generalization of label DP, it will also be valid for auditing a substitute DP mechanism, even though it might not be optimal for that. While DP-SGD with substitute accounting is a valid label DP mechanism, in practice, label DP is implemented using very different methods. As such, our auditing techniques would not be suitable for those methods. We make note of this in Section 7 (Discussion and Conclusion).
>
> [3] Chaudhuri, K., and Hsu, D. J. Sample Complexity Bounds for Differentially Private Learning. COLT 2011.

---

### Meta-Review · Area_Chair_ZVTL · 2025-12-28

**Summary:**

All reviewers agreed with the importance and novelty of the work. The authors addressed almost all concerns. As I do not see any outstanding concerns remaining, I recommend accepting this paper.

**Reviewer Concerns:**

* Reviewer hf2R:
  * The authors have addressed almost all concerns raised by the reviewer. In my opinion, especially the reviewer’s question/suggestion about the attacker’s prior knowledge was a good point and Table 2 is a good addition per the request/review.
Having said that, regarding computational cost, especially its scalability to larger applications, the authors did not explicitly respond to it. The effect of the number of training runs (Figure 4) is not translated to the scalability to larger applications. Upon acceptance of this paper, the authors are expected to discuss this issue and update the manuscript.

* Reviewer GPRK:
  * The authors addressed all concerns of the reviewer.

* Reviewer Hu7w:
  * The authors addressed all concerns of the reviewer.

* Reviewer tuq2:
  * Regarding the concern about generalizability to other gradient-based methods, the authors’ response was hypothetical, lacking supporting details or evidence.

**Reviewer Scores:**

* Reviewer hf2R: I think the reviewer would have increased the score.

* Reviewer GPRK: I do not think the reviewer would have increased the score.

* Reviewer Hu7w: I do not think the reviewer would have increased the score.

* Reviewer tuq2: I do not think the reviewer would have increased the score.

---

### Decision · Program_Chairs · 2026-01-26

Accept (Poster)